Differences in subjective well-being between individuals with distinct Joint Personality (temperament-character) networks in a Bulgarian sample

Garcia Danilo danilo.garcia@icloud.com 1 2 3 4 5
Kazemitabar Maryam 6 7
Stoyanova Kristina 8
Stoyanov Drozdstoy drozdstoy.stoyanov@mu-plovdiv.bg 8 9
Cloninger C. Robert 3 7 10
1 Centre for Ethics, Law and Mental Health (CELAM), University of Gothenburg , Gothenburg , Sweden
2 Department of Psychology, Lund University , Lund , Sweden
3 Department of Psychology, University of Gothenburg , Gothenburg , Sweden
4 Department of Behavioral Sciences and Learning, Linköping University , Linköping , Sweden
5 Promotion of Health and Innovation (PHI) Lab, International Network for Well-Being , Sweden
6 Yale School of Public Health , New Haven , Connecticut, USA
7 Promotion of Health and Innovation (PHI) Lab, International Network for Well-Being , USA
8 Research Institute at Medical University, Medical University of Plovdiv , Plovdiv , Bulgaria
9 Department of Psychiatry and Medical Psychology, Medical University of Plovdiv , Plovdiv , Bulgaria
10 Department of Psychiatry, Washington University School of Medicine , St. Louis , Missouri, USA
Greco Gianpiero
Electronic publication date: 2022 Aug 26
Publication date: 2022
Volume: 10
Electronic Location ID: e13956
Received 2022 May 16; Accepted 2022 Aug 6
Copyright: ©2022 Garcia et al.
Copyright year: 2022
Copyright holder: Garcia et al.
License: This is an open access article distributed under the terms of the Creative Commons Attribution License, which permits unrestricted use, distribution, reproduction and adaptation in any medium and for any purpose provided that it is properly attributed. For attribution, the original author(s), title, publication source (PeerJ) and either DOI or URL of the article must be cited.
License URL: https://creativecommons.org/licenses/by/4.0/

Keywords: Personality profiles, Temperament, Character, Affectivity, Life satisfaction, Subjective well-being, Bulgaria, Latent class analysis, Latent profile analysis, Joint personality networks

Funding: The authors received no funding for this work.

==============================
Background

Personality is the major predictor of people’s subjective well-being (i.e., positive affect, negative affect, and life satisfaction). Recent research in countries with high-income and strong self-transcendent values shows that well-being depends on multidimensional configurations of temperament and character traits (i.e., Joint Personality Networks) that regulate the way people learn to adapt their habits to be in accord with their goals and values, rather than individual traits. To evaluate the prevalence and the associations of different Joint Personality (temperament-character) Networks with well-being in a low-income country with weak self-transcendent values, we tested their association in Bulgarian adults, a population known to have strong secular-rationalist values but weak self-transcendent values.

Method

The sample consisted of 443 individuals from Bulgaria (68.70% females) with a mean age of 34 years (SD = 15.05). Participants self-reported personality (Temperament and Character Inventory), affect (Positive Affect Negative Affect Schedule), and life satisfaction (Satisfaction with Life Scale). The personality scores were used for profiling through latent profile analysis and latent class analysis based on temperament configurations (i.e., Temperament Profiles) of high/low scores of Novelty Seeking (N/n), Harm Avoidance (H/h), Reward Dependence (R/r), and Persistence (P/s); and character configurations (i.e., Character Profiles) of high/low scores of Self-Directedness (S/s), Cooperativeness (C/c), and Self-Transcendence (T/t).

Results

We found two Temperament Profiles and two Character Profiles that clustered into two distinctive Joint Personality Networks. All individuals in Joint Personality Network 1 had a Reliable (nhRP) Temperament Profile in combination with an Organized (SCt) Character Profile (i.e., a stable temperament and a healthy character configuration). About 71.9% in Joint Personality Network 2 had an Apathetic (sct) Character Profile in combination with Methodical (nHrp) or Reliable (nhRP) Temperament Profiles, while 28.1% had a Methodical (nHrp) Temperament Profile in combination with an Organized (SCt) Character Profile. Few people with high self-expressive values (i.e., high in all three character traits; SCT) were found. Individuals with a Joint Personality Network 1 with strong secular-rationalist values reported higher levels of positive affect and life satisfaction (p < .001), while individuals with a Joint Personality Network 2 reported higher levels of negative affect (p < .001).

Conclusions

Although a stable temperament and a healthy character were separately important for well-being, it was clear that it was the interaction between such temperament and character configuration that yielded greater levels of subjective well-being. Nevertheless, future research needs to investigate this interaction further to evaluate other cultures with variable configurations of personality traits and values.

Introduction

Personality is the major factor and predictor of people’s subjective well-being. Some researchers have argued that it is so because personality is related to the way people emotionally react (e.g., how intensively, duration) to life experiences (Kim-Prieto et al., 2005). However, personality is more than emotional responses to life events or temperamental dispositions, which do not account for environmental learning experiences regulated by our character (i.e., our goals and values) (Cloninger, 2004). Consideration of only a person’s temperament limits the concept of personality to traits that are emotion-based and moderately stable (McAdams, 2001). Instead, the science of human well-being (Cloninger, 2004; Cloninger & Cloninger, 2020) needs to: (a) account for both between- and within-individual differences in nonintentional (i.e., temperament) and intentional (i.e., character) domains of personality (Cervone, 2005)—people do not only differ between each other, which describes how people generally are in relation to others; but also differ within themselves in the way their temperament and character traits are organized, as we need to understand in order to predict how and why people behave as they do; (b) consider the large evidence of intraindividual variability across personality profiles (Ryan & Sackett, 2012), (c) account for the dynamics of personality development as a set of learning systems that adapt in a predictable and integrative manner over time (Cloninger, Svrakic & Svrakic, 1997; Zwir et al., 2020a; Zwir et al., 2020b; Zwir et al., 2022), and (d) consider recent molecular studies showing that the basic unit of personality are multidimensional profiles of temperament and character, not single traits (Cloninger & Zwir, 2018; Zwir et al., 2020a; Zwir et al., 2020b; Zwir et al., 2022).

In this context, Cloninger’s biopsychosocial model (Cloninger, Svrakic & Przybeck, 1993) decomposes personality in two domains comprised of seven personality dimensions that are based on robust research on the differences in the major brain systems for procedural versus propositional learning. According to Cloninger (2004), the temperament domain reflects the basic organization of independently different brain systems for the activation, maintenance, and inhibition of behavior in response to stimuli. The four temperament dimensions are defined in terms of individual differences in behavioral learning mechanisms shared by all animals, explaining responses to novelty and signals of reward or relief of punishment (Novelty Seeking), responses to signals of punishment or non-reward (Harm Avoidance), responses to social and attachment rewards (Reward Dependence), and the maintained response to previously rewarded behavior with intermittent reinforcement (Persistence). In contrast, the character domain involves individual differences in self-concepts about goals and values (Cloninger, 2004), which depend on brain systems that developed later in evolution (Cloninger, 2009; Zwir et al., 2022). Character is comprised of three dimensions: Self-Directedness (based on the concept of the self as an autonomous individual) allows the individual to engage in purposeful actions because the individual has a “sense of following a meaningful direction in one’s life” (Cloninger, 2004, p. 120); Cooperativeness (based on the concept of the social self) allows the individual to be tolerant and flexible about choices regarding goals because thought and behavior are based on mutual interests with other persons; and Self-Transcendence (based on the concept of the self with values derived from awareness of being an integral aspect of a larger whole, such as humanity, nature, and possibly the universe and its source) allows the individual to intuitively recognize the values and meaning in all things (see Table 1 for a detailed description of the different personality dimensions). In short, character allows us to act intentionally and interpret the meaning of what we experience, which in turn allows us to self-regulate our emotional reactions and even our habits (Cloninger, 2004; Moreira, Inman & Cloninger, 2021a). Due to its distinction between nonintentional (i.e., temperament) and intentional (i.e., character) domains of personality, Cloninger’s biopsychosocial model is appropriate for assessment of both within-person learning processes and between-person differences (Cervone, 2005)—that is, the way people differ from others but also the processes that motivate and regulate adaptive processes occurring within the individual.

Table 1 Descriptors of high and low scorers on the Temperament and Character Inventory (TCI) subscales.

Personality domain	TCI scales	TCI subscales	High scorers	Low scorers	
TEMPERAMENT	Novelty Seeking	NS1 excitability	exploratory	reserved	
NS2 impulsivity	impulsive	rigid	
NS3 extravagance	extravagant	thrift	
NS4 disorderly	rule-breaking	orderly	
				
Harm Avoidance	HA1 pessimism	pessimistic	optimistic	
HA2 fearfulness	fearful	risk-taking	
HA3 shyness	shy	outgoing	
HA4 fatigability	fatigable	vigorous	
				
Reward Dependence	RD1 sentimentality	sentimental	objective	
RD2 openness	warm	aloof	
RD3 attachment	friendly	detached	
RD4 dependent	approval-seeking	independent	
				
Persistence	PS1 eagerness	enthusiastic	hesitant	
PS2 hard-working	determined	spoiled	
PS3 ambition	ambitious	underachieving	
PS4 perfectionism	perfectionistic	pragmatic	
				
CHARACTER	Self-Directedness	SD1 responsibility	responsible	blaming	
SD2 purposefulness	purposeful	aimless	
SD3 resourcefulness	resourceful	helpless	
SD4 self-acceptance	unpretentious	pretentious	
SD5 self-actualizing	self-actualizing	unfulfilled	
				
Cooperativeness	CO1 social tolerance	tolerant	prejudiced	
CO2 empathy	empathetic	self-centered	
CO3 helpfulness	considerate	hostile	
CO4 compassion	forgiving	revengeful	
CO5 conscience	principled	opportunistic	
				
Self-Transcendence	ST1 self-forgetfulness	engaged	self-concerned	
ST2 transpersonal identification	joyfully connected, altruistic	separate individualistic	
ST3 spiritual acceptance	faithful	skeptical	
ST4 contemplation	contemplative	conventional	
ST5 idealism	idealistic	cynical	
Notes.

Adapted with permission from Anthropedia Foundation.

NS Novelty Seeking

HA Harm Avoidance

RD Reward Dependence

PS Persistence

SD Self-Directedness

CO Cooperativeness

ST Self-Transcendence

Hence, when individuals are asked to assess their subjective well-being, the recollection of a happy life is not exclusively and unconsciously dictated by how their temperament leads them to emotionally react (Cloninger, 2004). In fact, our recent genomic research shows that most of the genes associated with character are long-non-coding RNA genes that regulate the expression of protein-coding genes, coordinate the co-expression of sets of genes, and influence epigenetic processes. In contrast, most of the genes associated with temperament are protein-coding genes involved in cellular processes of synaptic plasticity, associative conditioning, and related processes of stress reactivity and neurotransmission (Zwir et al., 2020a; Zwir et al., 2020b; Zwir et al., 2022). Moreover, the genes encoding human character are associated with one brain network for higher cognitive processes involving intentional self-control and another brain network for self-awareness, whereas the genes encoding human temperament are enriched in highly conserved molecular pathways that are present in all animals and that are activated in experimental animals by associative conditioning in response to extracellular stimuli (Cloninger & Cloninger, 2020). In other words, despite the fact that human personality is moderately heritable (e.g., Gillespie et al., 2003; Ando et al., 2004; Garcia et al., 2013), the path to well-being and a resilient life depends on processes of learning, development, and integration of character development, such as self-actualization and identity formation that are optimized by the self-awareness of human beings, which allows the unique capacities of human self-aware consciousness (Cloninger & Cloninger, 2020). Put in another way, we inherit the way we learn, so nature and nurture are both always important.

We have replicated these molecular findings in three large independent genome-wide association studies from Finland, Germany, and South Korea (Zwir et al., 2020a; Zwir et al., 2020b; Zwir et al., 2022). Moreover, in these three independent samples, we uncovered three clusters of similar numbers of people with distinct combinations of Temperament and Character Profiles, which we refer to as Joint Personality (temperament-character) Networks. In short, temperament and character traits are expressions of the activity of three genetic-environmental networks that regulate healthy longevity and dissociable systems of learning and memory by nearly disjoint sets of genetic and environmental influences. Indeed, since personality is a complex adaptive system or a whole-system unit, it should be best studied by analyzing patterns of information rather than single traits (Cloninger, Svrakic & Svrakic, 1997; see also Bergman & Wångby, 2014).

These three Joint Personality Networks were subsequently confirmed in a Portuguese sample of adolescents, where individuals with a stable or reliable Temperament Profile (low in Novelty Seeking, low in Harm Avoidance, high in Reward Dependence, and high in Persistence) in conjunction with a healthy or Creative Character Profile (high in all three character dimensions) reported fewer clinical problems and greater engagement with school (Moreira, Inman & Cloninger, 2021b; Moreira et al., 2021). Among adults, individuals with a Creative (high in all three character traits) or an Organized Character Profile (high in Self-Directedness and Cooperativeness but low in Self-Transcendence) consistently report the highest levels of well-being, healthy longevity, optimal cardiovascular health, including healthy lifestyle as well as reduced risk for chronic diseases (Cloninger, 2004). Having a Creative Character Profile is also linked with better heart rate variability or vagal tone in 24-hour recordings of heart rhythms (Zohar, Cloninger & Mccraty, 2013). To the best of our knowledge, the three Joint Personality Networks have been replicated in our molecular studies (Zwir et al., 2020a; Zwir et al., 2020b; Zwir et al., 2022) and the Portuguese study (Moreira, Inman & Cloninger, 2021b). In addition, these Joint Personality (temperament-character) Networks closely resemble groups identified in large-scale longitudinal studies of social values: cultural creatives (i.e., post-materialists with strong self-expressive, prosocial, and self-transcendent values corresponding to those with Reliable-Creative Personality Networks), materialists (i.e., with secular-rationalist values corresponding to those with Reliable-Organized Personality Networks), and traditionalists (i.e., those whose values and behavior depend mainly on authority-dependent conventions and habits corresponding to those with temperaments weakly regulated by character) (Ray & Anderson, 2000; Inglehart, 2018a). We initially identified the three networks in countries with different cultural values and environmental conditions (Finland, Germany, and South Korea). We have replicated our findings in a lower income country (Portugal), but recognize that there is a need to examine other cultures, such as Bulgaria, which has been shown in the World Values Survey to have strong secular-rationalist values typical of egocentric people with a Reliable-Organized Personality Network, but have weak self-expressive, prosocial, and self-transcendent values, which is unlike people in the Reliable-Creative Personality Network (Inglehart, 2018a). The levels of well-being in groups of people with materialist or secular-rationalist values are intermediate to those of people with creative cultural values and those with traditional values (Zwir et al., 2022; Inglehart, 2018a; Inglehart, 2018b), so Bulgaria represents an interesting contrast to other countries in which we have assessed the associations between well-being and Joint Personality Networks.

In this line of thinking, we investigated the prevalence of different Temperament and Character profiles and Joint Personality (temperament-character) Networks and differences in subjective well-being (i.e., positive affect, negative affect, and life satisfaction) between individuals in a population of Bulgarian adults. Importantly, since a culture’s distinctive values are often a product of its history, our Bulgarian sample is phenomenologically relevant to the Balkan’s history of repeated colonization, which might have ingrained people in Bulgaria with the specific capacity for balancing different and even conflicting values—“Balkan people survived colonization by learning to live ‘at the edge of compromise’ between their own values and the values of their colonizers”, that is, Balkan pluralism (Stoyanov & Fulford, 2021, pp. 171). In fact, as mentioned, Bulgarians have been found to have strong secular-rationalist values and weak self-expressive values in the World Values Survey (Inglehart, 2018a; Inglehart, 2018b) so we hypothesized that people with Reliable-Organized Personality Networks would be frequent whereas those with Reliable-Creative Personality Networks would be few in number. In other words, we expected that our Bulgarian sample would be characterized by a very self-directed and cooperative but pragmatic and skeptical outlook on the world (i.e., high Self-Directedness, high Cooperativeness, and low Self-Transcendence).

Method

Ethical statement

The study was approved by the National Ethics Committee of the Bulgarian Association of Health Care Professionals (Protocol No. 2/10.05.2021).

Participants

The sample consisted of 443 individuals from Bulgaria (age range 18 to 65; about 68.70% females) with a mean age of 34 years (SD = 15.05). Subjects provided verbal consent to participate in the study. In contrast to the linear analyses originally conducted with the same data (see Angelova, 2020), we used person-oriented analyses by first clustering individuals in distinct Temperament Profiles and Character Profiles (see the Supplementary Material for details). Second, we combined individuals’ Temperament and Character Profiles to cluster them in Joint Personality (temperament-character) Networks that represent personality as a complex adaptive system.

Measures

Personality

We used the validated official Bulgarian version (Tilov et al., 2012) of the Temperament and Character Inventory (Cloninger, Svrakic & Przybeck, 1993) to measure the four temperament traits and the three character traits in Cloninger’s biopsychosocial model of personality: Novelty Seeking (e.g., “I often try new things just for fun or thrills, even if most people think it is a waste of time”), Harm Avoidance (e.g., “I often feel tense and worried in unfamiliar situations, even when others feel there is little to worry about”), Reward Dependence (e.g., “I like to discuss my experiences and feelings openly with friends instead of keeping them to myself”), Persistence (e.g., “I often push myself to the point of exhaustion or try to do more than I really can”), Self-Directedness (e.g., “In most situations my natural responses are based on good habits that I have developed”), Cooperativeness (e.g., “I often consider other persons’ feelings as much as my own”), and Self-Transcendence (e.g., “I sometimes feel so connected to nature that everything seems to be part of one living organism”). The version used here contains 140 items using a 5-point Likert scale (1 = strongly disagree, 5 = strongly agree) and had good reliability with the following Cronbach’s alphas: .64 for Novelty Seeking, .84 for Harm Avoidance, .75 for Reward Dependence, .89 for Persistence, .86 for Self-Directedness, .81 for Cooperativeness, and .81 for Self-Transcendence (Angelova, 2020).

Subjective well-being

We used the Positive Affect Negative Affect Schedule–Short Form (Watson, Clark & Tellegen, 1988) to operationalize the affective component of subjective well-being. This is a 20-item scale designed to measure positive affect and negative affect as independent dimensions. Participants are instructed to rate to what extent they have experienced 10 positive (e.g., strong, proud, interested) and 10 negative emotions (e.g., afraid, ashamed, nervous) during the last weeks, using a five-point Likert scale (1 = very slightly, 5 = extremely). In the present study, the positive affect scale showed a Cronbach’s alpha of .85 and the negative affect scale showed a Cronbach’s alpha of .88 (Angelova, 2020).

Moreover, we used the Satisfaction with Life Scale (Diener et al., 1985) to operationalize the cognitive component of subjective well-being. This scale has five statements (e.g., “In most ways my life is close to my ideal”) that respondents are asked to rate their level of agreement to using a seven-point Likert scale (1 = strongly disagree, 7 = strongly agree). The Satisfaction with Life Scale showed a Cronbach’s alpha of .83 in the present study (Angelova, 2020).

Hence, subjective well-being was operationalized as composed of three individual variables: positive affect, negative affect, and life satisfaction (cf. Diener, 1984). Each subjective well-being variable was calculated using the mean of total scores of each of the scales.

Statistical procedure

For the first phase, explorative analyses, we calculated zero-order correlations between the temperament and character dimensions and the subjective well-being dimensions (see also Angelova, 2020). For the second phase, we used latent profile analysis (LPA) to identify and cluster the study sample into (a) Temperament Profiles, and (b) Character Profiles (see the detailed procedure in the Supplementary Material). These models were estimated using standardized mean scores for each of the four temperament and three character dimensions, respectively (continuous variables). For the third phase, our main set of analyses, we used latent class analysis (LCA) to cluster individuals into Joint Personality (temperament-character) Networks. This model was estimated by combining participants’ assigned Temperament Profiles and Character Profiles (categorical variables). For both the LPA and LCA, we determined the optimum number of latent profiles or networks by comparing the fit of a series of models with increasing numbers of profiles. Model fit was compared using the Akaike Information Criterion (Akaike, 1974), Bayesian Information Criterion (BIC; Schwarz, 1978), sample-size adjusted BIC (Sclove, 1987), and entropy (Celeux & Soromenho, 1996). For LPA, we also used an Analytic Hierarchy Process (Akogul & Erisoglu, 2017) to help determine the optimal number of profiles. In both the second and third phases, using standardized scores for all measures (z-scores), we conducted a series of MANOVA:s to test differences in personality and subjective well-being (for the full analyses of the second phase, Temperament Profiles and Character Profiles, please see the Supplementary Material). The use of z-scores allowed us to compare the relevant variables (personality dimensions or subjective well-being constructs) within each profile or network (e.g., to test if Harm Avoidance among individuals with a specific profile differs from their own levels of Novelty Seeking, if positive affect among individuals with a specific profile differs from their own levels of life satisfaction, and etcetera).

Results

Phase 1: Correlations between personality traits and subjective well-being

Table 2 displays the zero-order correlations between temperament and character traits and subjective well-being. Regarding Temperament, as expected, Harm Avoidance was negatively associated to positive affect (r = -.40, p < .001) and life satisfaction (r = -.29, p < .001), but positively related to negative affect (r = .42, p < .001); and Persistence was positively related to positive affect (r = .56, p < .001) and life satisfaction (r = .31, p < .001). Regarding Character, also as expected, Self-Directedness was positively associated to positive affect (r = .37, p < .001) and life satisfaction (r = .39, p < .001), and negatively related to negative affect (r = -.47, p < .001); Cooperativeness was also positively associated to positive affect (r = .37, p < .001) and life satisfaction (r = .39, p < .001), and negatively related to negative affect (r = -.47, p < .001); and Self-Transcendence was positively associated to positive affect (r = .32, p < .001). The lowest correlation, between personality traits and subjective well-being constructs, was that between Novelty Seeking and negative affect (r = .01, p = .882). See also (Angelova, 2020).

Table 2 Correlations between temperament traits, character traits, and subjective well-being constructs (i.e., positive affect, negative affect, and life satisfaction).

	Dimensions	NS	HA	RD	PS	SD	CO	ST	PA	NA	LS	
Temperament	Novelty Seeking (NS)											
Harm Avoidance (HA)	−.22**										
Reward Dependence (RD)	−.02	.01									
Persistence (PS)	−.13**	−.37**	.18**								
Character	Self-Directedness (SD)	−.22**	−.55**	.17**	.39**							
Cooperativeness (CO)	−.21**	−.24**	.44**	.31**	.46**						
Self-Transcendence (ST)	.10*	−.12*	.15**	.34**	−.03	.21**					
Subjective Well-Being	Positive Affect (PA)	.04	−.40**	.13**	.56**	.37**	.27**	.32**				
	Negative Affect (NA)	.01	.42**	−.03	−.15**	−.47**	−.25**	.06	.01			
	Satisfaction with Life (LS)	−.01	−.29**	.11*	.31**	.39**	.21**	.19**	.41**	−.30**		
Notes.

Highlighted cells are correlations above .20, which is the recommended minimum effect size representing a practically significant effect for social science data according to Ferguson, 2009). Bold: correlations between temperament and character dimensions; Underlined : correlations between temperament and subjective well-being constructs; Italics: correlations between character and subjective well-being constructs.

** <.001.

Phase 2: Prevalence of temperament profiles and character profiles and differences in subjective well-being

Before conducting the latent class analyses (LCA) in phase three of our study, we calculated Temperament Profiles and Character Profiles (phase 2) separately using latent profile analyses (LPA). The LPA revealed two Temperament Profiles (profile 1 which included 18.2% of the participants and profile 2 which included 81.8% of the participants) and two Character Profiles (profile 1 which consisted of 23.1% of participants and profile 2 with 76.9% of the participants). For more details, please see Supplementary Materials, here we only summarize the results and derived conclusions.

Individuals in both Temperament Profiles reported low levels of Novelty Seeking (n = low Novelty Seeking) and were symmetrically different regarding high/low Harm Avoidance (H = high Harm Avoidance/h = low Harm Avoidance), high/low Reward Dependence (R = high Reward Dependence/r = low Reward Dependence), and high/low Persistence (P = high Persistence/p = low persistence). We labeled Temperament Profile 1 Methodical because individuals with this profile are highly cautious due to high Harm Avoidance (H), orderly due to low Novelty Seeking (n), and objective due to the combination of high Harm Avoidance and low Reward Dependence (Hr). Hence suggesting that individuals with the Methodical (nHrp) Temperament Profile might be described as inhibited (nH = low Novelty Seeking and high Harm Avoidance), aloof (Hr = High Harm Avoidance and low Reward Dependence), privacy-seeking (nr = low Novelty Seeking and Low Reward Dependence), and having difficulties to initiate anything new because of their inhibitions rooted in their tendency to pragmatism and underachievement (p = low Persistence). If such an individual lacks a well-developed Character Profile, they can be perceived and act as obsessional personalities and find situations that require exposure to public attention to be challenging (Cloninger, 2004). They are, however, not afraid of being rejected (Hr = high Harm Avoidance and low Reward Dependence), hence, making them objective. Conversely, we labeled Temperament Profile 2 Reliable because individuals with this Temperament Profile are stable due to low Novelty Seeking and low Harm Avoidance (nh), warmly sociable due to low Harm Avoidance and high Reward Dependence (hR), traditional because of their low Novelty Seeking and high Reward Dependence (nR), and hard-working due to high Persistence (P). Hence, it is highly likely that individuals with a Reliable (nhRP) Temperament Profile can be trusted to carry out what they are expected to do in a predictable and traditional manner and to develop a mature character (Cloninger, 2004). As expected, a post hoc test with Bonferroni correction (see Supplementary Material) showed that positive affect and life satisfaction were higher among individuals with the Reliable (nhRP) Temperament Profile and negative affect was higher among individuals with the Methodical (nHrp) Temperament Profile (p < .001).

Individuals in both Character Profiles reported low levels of Self-Transcendence (t = low Self-Transcendence) but symmetrically different levels of high/low Self-Directedness (S = high Self-Directedness/s = low Self-Directedness) and high/low Cooperativeness (C = high Cooperativeness/c = low Cooperativeness). We labeled Character Profile 1 as Apathetic because individuals with this profile tend to feel victimized and helpless (sc = low Self-Directedness and low Cooperativeness), show very poor judgement (st = low Self-Directedness and low Self-Transcendence), and are distrustful (ct = low Cooperativeness and low Self-Transcendence). Indeed, individuals with an Apathetic (sct = low in all three character traits) Character Profile report the lowest levels of overall well-being and health, report experiencing unhealthy emotions such as anxiety and alienation, and have high rates of mental and physical disorders (Cloninger, 2004). In other words, they experience the world from an outlook of separateness, which leads to fear, excessive desire, and false pride or self-reproach. Conversely, we labeled Character Profile 2 Organized because individuals with such profile are often perceived as mature leaders (SC = high Self-Directedness and high Cooperativeness), logical (St = high Self-Directedness and low Self-Transcendence), and conventional (Ct = high Cooperativeness and low Self-Transcendence). They are, most of the time, happy and healthy, and seldom need health care (Cloninger, 2004). However, when they face difficult existential challenges, such as severe illness or death, they often lack the necessary outlook of unity and connectedness needed to be resilient through such situations due to low levels of Self-Transcendence (t). As expected, a post hoc test with Bonferroni correction (see Supplementary Material) showed that life satisfaction was higher among individuals with the Organized (SCt) Profile and negative affect was higher among individuals with the Apathetic (sct) Profile (p < .001). Positive affect, however, did not differ between individuals with these two Character Profiles. Thus, accentuating that an Organized (SCt) Profile is necessary, but not sufficient for experiencing a happy life.

Phase 3: Joint personality (temperament-character) networks

In phase 3, our main set of analyses, we conducted a LCA to investigate the interaction of the distinct Temperament Profiles and Character Profiles as Joint Personality (temperament-character) Networks. We tested four different models with one and up to four networks (Table 3). All values for Model 2, with two networks, had the best fit to our model (AIC = 840.770, BIC = 861.237, SABIC = 845.370, VLMRT = .0006, LMRT = .0008, and BLRT = <.001). Model 2 consisted of Joint Personality Network 1 which included 68.6% of the participants and Joint Personality Network 2 with 31.4% of the participants (see more details in the Supplementary Material).

Table 3 Latent class analysis for Joint Personality (temperament-character) Networks.

Model	AIC	BIC	SABIC	Entropy	VLMRT	LMRT	BLRT	
1	846.638	854.825	848.478					
2	840.770*	861.237*	845.370*	0.270	.0006*	.0008*	<.0001*	
3	846.770	879.518	854.130	0.737	.5131	.5131	1.0000	
4	852.770	897.799	862.890	0.845*	.5017	.5017	1.0000	
Notes.

* Optimum values for fit indices. The model number also indicates the number of networks within each model.

Regarding Temperament Profiles, all the individuals in the Joint Personality Network 1 had a Reliable (nhRP) Temperament Profile. In Joint Personality Network 2, as much as 46.8% of the individuals had a Methodical (nHrp) Temperament Profile and 53.2% had a Reliable (nhRP) Temperament Profile. In other words, the number of individuals with the Methodical (nHrp) Temperament Profile and the Reliable (nhRP) Temperament Profile were almost equal in the Joint Personality Network 2; while all individuals allocated to Joint Personality Network 1 had a Reliable (nhRP) Temperament Profile (see Table 4). Regarding Character Profiles, all the individuals in Joint Personality Network 1 had an Organized (SCt) Character Profile. Conversely, 28.1% of the individuals clustered in the Joint Personality Network 2 had an Organized (SCt) Character Profile and 71.9% had an apathetic (sct) Character Profile. This means that individuals allocated in the Joint Personality Network 1 had a significantly higher amount of individuals with an Organized (SCt) Character Profile compared to individuals in the Joint Personality Network 2. In sum, while all individuals in the Joint Personality Network 1 had a stable Reliable (nhRP) Temperament Profile in combination with a healthy Organized (SCt) Character Profile, none of the individuals in Joint Personality Network 2 had this stable and healthy personality configuration; instead 71.9% had an Apathetic (sct) Character Profile in combination with Methodical (nHrp) or Reliable (nhRP) Temperament Profiles and the rest (28.1%) had an Organized Character Profile in combination with a Methodical (nHrp) Temperament Profile (see the Supplementary Material for the details).

Table 4 Prevalence of individuals with different Temperament Profiles and Character Profiles clustered in each of the Joint Personality (temperament-character) Networks.

	Profiles	Joint Personality Network 1	Joint Personality Network 2	Total	
		n	%	n	%	N (%)	
Temperament Profiles	Methodical (nHrp)	0	0%	65	46.8%	65 (14.7%)	
Reliable (nhRP)	304	100%	74	53.2%	378 (85.3%)	
Total	304	100%	139	100%	443 (100%)	
Character Profiles	Apathetic (sct)	0	0%	100	71.9%	100 (22.6%)	
Organized (SCt)	304	100%	39	28.1%	343 (77.4%)	
Total	304	100%	139	100%	443 (100%)	
Notes.

r low Reward Dependence

n low Novelty Seeking

H high Harm Avoidance

h low Harm Avoidance

R high Reward Dependence

P high Persistence

p low persistence

S high Self-Directedness

s low Self-Directedness

C high Cooperativeness

c low Cooperativeness

t low Self-Transcendence

Differences in temperament and character dimensions within individuals with distinct joint personality (temperament-character) networks

We found significant differences in personality dimensions within each Joint Personality Network with a Greenhouse-Geisser correction (F(4.78,1447,78) = 9.12, p < .001, η2p = 0.03, observed power = 1.0). However, a Bonferroni adjustment test showed that some mean differences were not significant (p > .05). Within Joint Personality Network 1, individuals scored highest in Persistence and lowest in Harm Avoidance. Within Joint Personality Network 2, individuals scored the highest in Harm Avoidance and the lowest in Persistence (see Fig. 1). In other words, individuals in Joint Personality Network 1 were driven by, for example, perfectionism, optimism, and risk-taking. Conversely, individuals in Joint Personality Network 2 were driven by, for example, pessimism, fear, shyness, pragmatism, and underachievement.

Differences in temperament and character dimensions between individuals with distinct joint personality (temperament-character) networks

The differences in personality dimensions between individuals with distinct Joint Personality Networks were significant (Wilks’ Lambda = 0.76, F(7,435) = 20.03, p < .001, observed power = 1.0). A Bonferroni adjustment test showed that the Joint Personality Networks differed significantly with regards to all temperament and character dimensions except for Novelty Seeking (p = .045). Reward Dependence, Persistence, Self-Directedness, Cooperativeness, and Self-Transcendence were higher in the Joint Personality Network 1 compared to the Joint Personality Network 2, while Harm Avoidance was higher in the Joint Personality Network 2 (see Fig. 1). Hence, the method (i.e., LCA) for allocating individuals to different networks depending on their temperament profile and character profile seems valid. Nevertheless, we did not find a significant variation regarding Novelty Seeking, most individuals scored low in this temperament trait. Hence, indicating that individuals in both Joint Personality Networks are reserved, rigid, prudent with their economy, and dislike disorderliness. Moreover, even though individuals in these two Joint Personality Networks differed in Self-Directedness and Cooperativeness, individuals in both networks scored low in Self-Transcendence. Thus, most individuals in this sample are self-concerned, individualistic, skeptical, conventional, and cynical.

Figure 1 Mean differences (z-scores) in temperament and character dimensions between and within Joint Personality (temperament-character) Network 1 and 2.

Differences in subjective well-being (positive affect, negative affect, and life satisfaction) within individuals with distinct joint personality (temperament-character) networks

In Joint Personality Network 1, the test of within-subject effects with Greenhouse-Geisser correction was significant (F(1.72,508.60) = 9.80, p < .001, η2p = 0.03). The pairwise comparison with Bonferroni adjustment showed that the difference between positive affect and life satisfaction was not significant (p = 1.000)—that is, positive affect and life satisfaction were equally high. Negative affect, however, was significantly lower than both positive affect and life satisfaction (p < .001). In other words, confirming that individuals in Joint Personality Network 1 experienced positive emotions more frequently and were more satisfied with their life in relation to their own experience of negative emotions (see Fig. 2).

Regarding Joint Personality Network 2, the test within subject effects with Greenhouse-Geisser correction was also significant (F(1.72,287.26) = 17.28, p < .001, η2p = 0.11). Again, there was no difference between positive affect and life satisfaction (p = 1.000)—that is, positive affect and life satisfaction were equally low. However, a Bonferroni post hoc adjustment test indicated that in contrast to the differences within the Joint Personality Network 1, negative affect was significantly higher than both positive affect and life satisfaction (p < .001) within individuals with the Joint Personality Network 2. Hence, confirming that individuals in Joint Personality Network 2 experienced negative emotions more frequently in relation to their own experience of positive emotions and evaluations of life satisfaction (see Fig. 2).

Figure 2 Mean differences (z-scores) in subjective well-being between and within Joint Personality (temperament-character) Network 1 and 2.

Differences in subjective well-being (positive affect, negative affect, and life satisfaction) between individuals with distinct joint personality (temperament-character) networks

The last one-way MANOVA showed that there were significant differences between individuals with distinct Joint Personality Networks (Wilks’ Lambda = 0.90, F(3,439) = 15.76, p < .001, η2p = 0.10). The test between-subject effects indicated that the differences in life satisfaction, positive affect, and negative affect between individuals in Joint Personality Networks 1 and 2 were significant (p < .001). A Bonferroni post hoc correction test showed that individuals with a Joint Personality Network 1 reported higher levels of positive affect and life satisfaction (p < .001), while individuals with a Joint Personality Network 2 reported higher levels of negative affect (p < .001). See Fig. 2.

Hence, although a stable temperament and healthy character were separately important for well-being, it was clear that it was the interaction between such temperament and character configuration what yielded greater levels of subjective well-being.

Discussion

In this study we investigated the prevalence of different Temperament and Character profiles and found two Joint Personality (temperament-character) Networks in our Bulgarian sample. We also found differences in subjective well-being across individuals with distinctive networks. The Joint Personality Networks incorporated two Temperament Profiles, Methodical (nHrp) and Reliable (nhRP), and two Character Profiles, Apathetic (sct) and Organized (SCt). All individuals in the Joint Personality Network 1 had a Reliable (nhRP) Temperament Profile and an Organized (SCt) Character Profile. They experienced positive affect to a greater extent and were more satisfied with their lives compared to individuals in Joint Personality Network 2. Within Joint Personality Network 2 individuals belonged to the following profiles: 46.8% had a Methodical (nHrp) Temperament Profile, 53.2% of them had a Reliable (nhRP) Temperament Profile, 28.1% of them had an Organized (SCt) Character Profile and 71.9% had an Apathetic (sct) Character Profile. Compared to individuals in Joint Personality Network 1, these individuals experienced negative affect to a greater extent and lower levels of positive affect and life satisfaction. Our results agree with studies showing that personality combinations are distinctively associated to individual differences in both affective and cognitive aspects of subjective well-being. For example, in a study among middle age New Zealanders (Spittlehouse et al., 2014), individuals with Character Profiles high in both or either Self-Directedness and Cooperativeness reported higher levels of well-being. Thus, implying that agentic and communal behavior, rather than self-transcendent behavior, is important for our well-being. However, in this same study, it was shown that self-transcendent values or practices (i.e., self-expressive values) contribute to well-being when agentic (i.e., high Self-Directedness) and communal values (i.e., high Cooperativeness) are not well developed (Spittlehouse et al., 2014). In another study among university students, the Creative (SCT) Character Profile was associated with the highest levels of life satisfaction, whereas the Apathetic (sct) Character Profile was associated with the lowest levels of life satisfaction (Park et al., 2015). Similar results have been found in Finland (Josefsson et al., 2011), Israel (Cloninger & Zohar, 2011), Sweden (Schütz, Archer & Garcia, 2013), and other countries (e.g., Giakoumaki et al., 2016; Wang, Hu & Li S. Tao, 2019).

For instance, a study in the Bulgarian army population led to similar results (Dimitrova, Ganev & Donchev, 2015). Clinical researchers showed that individuals who were patients diagnosed with personality disorders reported low Novelty Seeking, low Persistence, and high Harm Avoidance (i.e., similar to the Methodical Temperament Profile in the present study) and low Cooperativeness, low Self-Directedness, and high Self-Transcendence. Conversely, Self-Transcendence was low in both Character Profiles we found in our study (i.e., the Apathetic Profile and the Organized Profile). On the other hand, healthy military servicemen reported high Novelty Seeking, high Persistence, and low Harm Avoidance; which is also in contrast to our study where Novelty Seeking in both Temperament Profiles was low (Dimitrova, Ganev & Donchev, 2015). Nevertheless, high Novelty Seeking among healthy military recruits seems reasonable (Mommersteeg et al., 2011). Moreover, healthy military servicemen reported high Self-Directedness, high Cooperativeness, and low Self-Transcendence (Dimitrova, Ganev & Donchev, 2015); which is similar to the Organized Character Profile in the present study. To the best of our knowledge, however, the present study is the first one to investigate Joint Personality (temperament-character) Networks in the Bulgarian population, rather than single traits, and one of the few overall using LPA and LCA, rather than median splits or other clustering methods, to replicate past molecular studies (Zwir et al., 2021). Indeed, LPA and LCA are data-driven and create profiles and networks that are relative to each other, which comes closer to modeling the dynamic nature of within and between group variability of individual patterns of temperament and character and their combination. What is even more, in contrast to other clustering algorithms, the methods used here allow for ”model-based clustering” using a probabilistic model that describes data distribution—that is, in contrast to the bottom-up approach of cluster analyses in which clustering is done by finding similarities between cases, LPA and LCA are top-down approaches in which clustering starts with describing data distribution and use a statistical model for data selection and assessment of goodness of fit (Hagenaars & McCutcheon, 2002).

In the present study, the Bulgarian participants were classified in two Joint Personality Networks that, besides Novelty Seeking and Self-Transcendence, were almost diametrically different in terms of temperament and character traits. The Joint Personality Network 1 is represented by a more consolidated cohort of people with a Reliable (nhRP) Temperament Profile and Organized (SCt) Character Profile, which describes them as individuals with a stable temperamental disposition and a more mature character. The Joint Personality Network 2 is more heterogeneous as it is represented by all temperament-character configurations but the one in Joint Personality Network 1 (i.e., Reliable-Organized). These findings suggest that, if individuals with a Reliable (nhRP) Temperament Profile, who reported higher levels of subjective well-being compared to those with a Methodical (nHrp) Temperament Profile, have an Apathetic (sct) Character profile; they will still end up with low levels of subjective well-being. Accordingly, if individuals with an Organized (SCt) Character Profile, who reported higher levels of subjective well-being compared to those with an Apathetic (sct) Character Profile, have a Methodical (nHrp) Temperament profile; they will still end up with low levels of subjective well-being. In other words, although a stable temperament and a healthy character were separately important for well-being, it was clear that it was the interaction between such temperament and character configuration what yielded greater levels of subjective well-being in this Bulgarian sample.

This conclusion is important because it goes beyond what can be inferred by just studying traits or specific dimensions of personality or even temperament profiles and character profiles separetly. Harm Avoidance for example, is a primary personality trait associated with restraint of behavior (Cloninger, 1987; Láng, 2020). In our Bulgarian sample this was confirmed by a significant association between high Harm avoidance and high negative affect, as well as the fact that individuals with a Methodical (nHrp) Temperament Profile reported higher levels of negative affect than those with a Reliable (nhRP) Temperament Profile. At first sight, this might indicate that low Harm Avoidance is determinant for low negative affect. However, it might only be necessary but not sufficient. After all, individuals with the configuration Reliable-Organized (i.e., Joint Personality Network 1) emerged as the ones with the lowest levels of negative affect, while those with a Reliable-Apathetic configuration, despite low levels of Harm Avoidance, reported higher levels of negative affect. That being said, in our study, we lacked a network representing cultural creatives, that is, those with a Reliable Temperament Profile and Creative (SCT = high in all character traits) Character Profile. Indeed, most of our Bulgarian population were low in Self-Transcendence. It is plausible to argue that a Creative Character Profile might always help the individual to regulate the emotional reactions and experiences from any type of Temperament Profile. Indeed the path to well-being and a resilient life depends on processes of learning, development, besides the integration of character development (Cloninger & Cloninger, 2020). Put in another way, we inherit the way we learn, so nature and nurture are both always important. Thus, to cope with high levels in Harm Avoidance, character development is extremely important, but targeting the nervous system is also necessary (see for example Cloninger et al., 2019).

The lack of a third Joint Personality Network, previously found in Finland, Germany, South Korea, and Portugal, is in fact our most significant finding. We had reasons for expecting such results. After all, our Bulgarian sample is phenomenologically relevant to the Balkan’s history of repeated colonization, which might have ingrained people in Bulgaria with the specific capacity for balancing different and even conflicting values, that is, Balkan pluralism (Stoyanov & Fulford, 2021). In fact, Bulgarians seem to have strong secular-rationalist values and weak self-expressive or self-transcendent values (Inglehart, 2018a; Inglehart, 2018b). Thus, people with a Reliable-Organized Network should be the most frequent configuration whereas those with a Reliable-Creative Network should be few. Furthermore, the absence of a Reliable-Creative Network is perhaps also related to the authoritarian history of Bulgaria. According to Inglehart (2018a) authoritarian systems that suppress self-expression and democracy tend to be individualistic and materialistic and show less development of the creative self-awareness system.

Limitations

In the present study we only had age and gender as demographic variables, education, for example, might be an important factor behind our results. Moreover, self-report scales might result in consciously or unconsciously biased accounts of individuals’ experiences and are also biased specifically by social desirability. Nevertheless, the ability of respondents to self-assess accurately is a limitation that self-report measures have in general.

Conclusions

Recent studies provide evidence for the relation between personality as a complex biopsychosocial adaptive system and well-being. These results reveal not only how people differ from each other but also how and why certain people are happier and more satisfied with their life than others. Our results are also an addition to the debate of how and why different cultures might differ (cf. Allik & McCrae, 2004) regarding the development of these Joint Personality (temperament-character) Networks. We argue that the biopsychosocial model of personality can capture the multi-dimensional complexity of subjective well-being in a variety of socio-cultural contexts. Importantly, adaptive traits can be cultivated to elevate one’s levels of well-being (Caspi, Roberts & Shiner, 2005; Cloninger et al., 2019; Cloninger & Cloninger, 2020). It is fundamental to reveal a broader spectrum and level of analysis to personality in order to provide interventions for personality development as well as a culture that allows individuals to strengthen their well-being by intergrating their cognition, emotions, and behavior. That is, a culture that supports cultural creatives (cf. Inglehart, 2018a; Inglehart, 2018b) and in that way supports individual and social resilience.

Supplemental Information

Supplemental Information 1 Bulgarian Personality

Click here for additional data file.

Supplemental Information 2 Supplemental Material

Click here for additional data file.

We would like to thank Sofia Angelova, who assisted with the data collection.

Additional Information and Declarations

Competing Interests

Author Contributions

Human Ethics

Data Availability

The authors declare there are no competing interests.

Danilo Garcia analyzed the data, prepared figures and/or tables, authored or reviewed drafts of the article, and approved the final draft.

Maryam Kazemitabar analyzed the data, prepared figures and/or tables, authored or reviewed drafts of the article, and approved the final draft.

Kristina Stoyanova analyzed the data, authored or reviewed drafts of the article, and approved the final draft.

Drozdstoy Stoyanov conceived and designed the experiments, performed the experiments, authored or reviewed drafts of the article, and approved the final draft.

C. Robert Cloninger conceived and designed the experiments, authored or reviewed drafts of the article, and approved the final draft.

The following information was supplied relating to ethical approvals (i.e., approving body and any reference numbers):

National Ethics Committee of the Bulgarian Association of Health Care Professionals.

The following information was supplied regarding data availability:

The raw data, all scales that are the subject of the performed analyzes, is available in the Supplemental File.

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
