# Peer review of "Differences in subjective well-being between individuals with distinct Joint Personality (temperament-character) networks in a Bulgarian sample"

_PeerJ, doi:10.7717/peerj.13956_

## Round 0.1 · original submission · Major Revisions

Dear Authors,

Respond point by point to reviewers' comments and follow their suggestions carefully.

Reviewer 1 ·

Basic reporting

Clear and unambiguous, professional English used throughout.

Experimental design

Rigorous investigation performed to a high technical & ethical standard.

Validity of the findings

All underlying data have been provided; they are robust, statistically sound,

Additional comments

Thanks for opportunity to review manuscript entitled '' Personality profiles in a Bulgarian sample: differences in Subjective Well-Being between Temperament and Character Profiles'' for Peerj Journal.
The article is well written and novel. I have some recommendations for this manuscript.
1. All p values are reported wrongly according to APA-7 rules. It is impossible to be exactly zero a p value. All p = .000 must be corrected as p < .001 along the manuscript.
2. Authors must add more information about study sample demographics such as education level, and other demographics relevant to study.
3. Authors must improve study importance. Specifically, authors need to answer why it is important to examine Personality profiles in a Bulgarian sample? with citations and other supporting information.
4. All figures and tables must re upload to main manuscript. Their image quality is very bad. I am not able to asses most of them.
5. How authors constructed subjective well-being is unclear. Are authors constructed as positive affect + life satisfaction-negative affect? I better explanation required for that part.
6. Practical implications of study findings is completely missing and must be added.
7. Some information in latent class analyses and latent profile analyses is missing. Authors reported statistics is not consistent with Tables. Authors must add statistical analyses in Table to Statistical analyses section.
8. Reporting of statistics along the manuscript is wrong along the manuscript as per APA 7 rules as well as reporting eta-square. I think withing subject Manova uses partial eta-square and checking this may be useful.

Reviewer 2 ·

Basic reporting

no comment

Experimental design

minor issue - they did not use a within-participants design and should not mis-describe a between-participants analysis as 'within-subjects'

Validity of the findings

no comment

Additional comments

This article applies Cloninger’s psychobiological model of personality to a sample of Bulgarian participants – organizing them into two empirically-based Joint Profiles and testing for individual differences in subjective well being and life satisfaction. It is an informative study with respect to the relationship between personality and health/pathology and also has some interesting findings regarding some potentially distinct features of the Bulgarian population compared to other populations.

I mostly have suggestions for clarifications to make the article a little easier to read.

The abstract refers to recent molecular research showing that the basic units of personality are profiles of temperament and character, but this was not discussed in the article. This refers to the Zwir articles in the reference list? The problem is that the ‘molecular’ basis of this is not explained, even briefly – and in truth it may not be that relevant to this study. Also, there is an authors’ note on line 67 to update the Zwir references with page numbers – that should be deleted.

The introduction says both between-participant and within-participant differences are important, but it does not look like within-participant differences are a part of this study, i.e., there is only a between-participants design, not a within-participants (repeated measures) design. If so, I would suggest not setting up an expectation that this article will address both.

On that note, what is mean by within-subjects effects in line 270 and following? This looks like for each Joint Profile separately, it is a between-participants analysis looking at differences between the variables for that group only.

For the zero order coefficients, what is more relevant are probably the correlations between the predictors (Cloninger’s constructs) and the criterion variables (affect and satisfaction). It might make more sense, if the important correlations are going to be summarized in the text, to states the highest and the lowest correlations between the predictors and the criteria. Not between the predictors.

The first time I read it, it was a little hard to recall the differences between Temperament Profiles, Character Profiles and Joint Personality Networks. Not on the second reading.

It also took me a little while to figure out some of the coding like nHrp = low novelty seeking, high harm avoidance, low reward dependence and low persistence. It still took some effort to follow that on a second reading. If there is space, rather than (ct) it might be better to write (low cooperativeness and low transcendence). I would also capitalize Joint Personality Network (or is it profile), Temperament Profile, and Character Profile in the text. It would also help to have a Table that gives the different combinations for the profiles before we get to Table 4. Even better, when introducing them start with “The first Temperament Profile is called Methodical,” then explain it. Then in a separate paragraph “The second Temperament Profile is called Reliable” - etc. Might be easier to track going forward and separate paragraphs easier to find if there is a need for a reader to flip back through.

It looks to me like Joint Profile 1 is more homogeneous, whereas Joint Profile 2 is a conglomeration. Would it not be the case that they key variable with respect to high negative affect in Joint Profile 2 is due to the Apathetic Character Profile alone as suggested by the New Zealand study (i.e., Joint Profile 2 is not needed or has limited incremental validity in addition to the Character Profile). This is where the molecular findings might be relevant to discuss further.

I did not follow the bit about Balkan pluralism. Why is this study relevant to that issue? Does it refer to contradictory roles in society or within persons? More could be said here as it seems interesting.
Minor
Alpha coefficients were reported for the PANAS and Diener’s scale but not Cloninger’s constructs. Why not? They should be easy enough to run.
This sentence does not make sense – line 58 However, personality is more than just habit
59 learning of nonintentional emotional responses or temperamental dispositions (

---

## Round 0.2 · accepted · Accept

Dear authors, you have done an excellent job revising the manuscript by carefully following the valuable suggestions of the reviewers.

Reviewer 1 ·

Basic reporting

Clear and unambiguous, professional English used throughout.

Experimental design

Method and result section is correct.

Validity of the findings

The findings are valid.

Additional comments

Thanks for opportunity review revised manuscript entitled ‘‘Differences in subjective well-being between individuals with distinct Joint Personality (temperament-character) Networks in a Bulgarian sample’’. I would like the thanks to authors. They make a good job for improving quality of their manuscript. Authors revised the manuscript as I requested with a good will. In this form, Introduction reflects very well the previous studies and study aim, Method section and Result section is correct, and Discussion section adequately synthesis to previous study findings and current study results. Overall, I have no further comment regarding to manuscript. I congratulate to authors and wish them success on their future endeavors.